# PDCSA: A parallel discrete crow search algorithm for influence maximization in social networks

**Lihong Han**[1,2]*, **Kan Yang**[1,2], **Yang Ming**[1,2], **Jianxin Tang**[3]

1 School of Statistics and Data Science, Lanzhou University of Finance and Economics, Lanzhou, China,
2 Key Laboratory of Digital Economy and Social Computing Science, Gansu Province, Lanzhou, China,
3 School of Computer and Communication, Lanzhou University of Technology, Lanzhou, China

* hanlh16@lzu.edu.cn

## Abstract

The essence of the influence maximization (IM) problem is how to identify the set of seed nodes so that the node numbers ultimately affected in the network reach the maximum under a certain spreading model. In the field of influence maximization research, the investigation of seed nodes identifying algorithms is a hot yet challenging work. Although conventional greedy algorithms and heuristic algorithms have high performance, their efficiency remains a challenge when applied to large-scale social networks. In recent years, swarm intelligence-based optimization algorithms have seen increasing application in addressing this problem, with notable improvements in performance. However, the efficiency of these swarm intelligence-based algorithms still needs to be improved in large-scale social networks. Based on this issue, a parallel discrete crow search algorithm (PDCSA) designed for parallel computing is proposed. Based on the evolution characteristics, PDCSA makes full use of the efficiency advantage of parallel computing to improve the time efficiency of solving IM problems. The results of experiments conducted on six datasets show that PDCSA achieves performance comparable to state-of-the-art algorithms, with the added advantages of high efficiency and robustness.

## Introduction

There are many types of network graphs presently, including supply chain networks [1], software-defined networks [2], social network [3], etc. Based on social media, social networks like TikTok and Twitter are ubiquitous, as they enable people to follow what they are interested in, share what they find interesting, and get closer to their friends. These networks have become an indispensable part of real social interaction and significantly influenced people's lives [4]. In today's complex information environment, individuals are increasingly inclined to accept information provided by well-known individuals or reputable organizations. This tendency can be attributed to the desire to reduce the cognitive effort and resources required to independently verify

**Data availability statement:** All relevant data are within the manuscript and its Supporting Information files.

**Funding:** This work was partially supported by the Gansu Sci&Tech Program under Grant No. 22JR11RA134, Gansu Provincial Fund for Distinguished Young Scholars under Grant No. 23JRRA766, National Social Science Fund of China under Grant No. 21BTJ042, Financial Statistics Research Integration Team of Lanzhou University of Finance and Economics under Grant No.XKKYRHTD202304. The funders had no role in study design, data collection and analysis, decision to publish, or preparation of the manuscript.

**Competing interests:** The authors have declared that no competing interests exist.

the accuracy of information. Similarly, when a new product is launched, a critical challenge arises: how to strategically select an initial group of users within a social network to maximize the product's word-of-mouth effect. Real-world problems of this nature can be abstracted into the Influence Maximization (IM) problem in social networks, which has a wide range of practical applications in marketing, information dissemination, and social behavior analysis. Since the initial proposal and formalization of the IM problem by Kempe *et al.* [5], many scholars have studied this problem and put forward many methods to solve it.

In this work, we propose a parallelized discrete crow search algorithm (PDCSA) to solve the problem of seed node set identification in large-scale social networks, which is appropriate for multi-threaded concurrent computing. In summary, the primary contribution of this work follows:

- Based on the network structure, the position vector and memory vector of the crow flock were encoded.

- An innovative search strategy that synergizes local search with global exploration through random walk-based diversification.

- The framework of Parallel Discrete Crow Search Algorithm was developed to enable efficient parallel computation for influence maximization in large-scale social networks.

The remainder of this article is structured as follows. The relevant research is reviewed in Related work section, while Preliminaries section depicts the preliminaries and definitions used in this investigation. The framework of PDCSA is presented in Proposed method section. The grid search strategy utilized to determine the optimal parameter values of the PDCSA for the IM problem, and the performance evaluation of PDCSA on six experimental networks, along with the analysis of the results, is presented in Experiments section. The concluding remarks and future work directions are presented in the last section.

## Related work

The IM problem is an NP-Hard problem [6]. Addressing this problem involves tackling two core challenges: the accurate evaluation of node influence and the effective reduction of influence overlap among selected nodes. To solve these problems, researchers have explored diverse approaches and proposed numerous strategies aimed at improving both the effectiveness and efficiency of IM in social networks.

### Centrality-based approaches

A common approach to addressing the first challenge involves utilizing various centrality index to quantify the influence of individual nodes, such as Degree Centrality [7], Betweenness Centrality [8], Closeness Centrality, Eigenvector Centrality, etc. Degree Centrality is determined by the number of edges connected to a node's neighbors. A higher degree centrality indicates that a node has more connections, and thus potentially exerts greater influence within the network. Betweenness

Centrality is based on the role a node plays in the shortest paths across the network. Specifically, it quantifies the proportion of all shortest paths between pairs of nodes that through the given node. This measure reflects the node's influence under the assumption that a higher proportion of such paths indicates a more significant bridging role, and thus greater potential for influence within the network. Many other network centrality characteristics are available for node influence evaluation that are similar to Degree Centrality and Betweenness Centrality.

## Heuristic algorithms

While node centrality-based methods are effective in identifying influential nodes, they generally fail to account for the overlapping influence among seed nodes. To address this limitation, scholars have proposed various optimization approaches aimed at mitigating the overlapping impact of influence between seed nodes. In the early stages of research on IM problem, Kempe *et al.* [5] proposed a greedy mechanism-based algorithm that demonstrated strong performance. This approach iteratively selects nodes that maximize the spread of influence by evaluating all nodes in the network. Obviously, the computational complexity of this method becomes high as network size increases, leading to severe efficiency issues. To address this limitation, CELF [9] and CELF++ [10] algorithms that based on the greedy mechanism are proposed to enhance the original greedy algorithm. Experiments demonstrate that the enhanced algorithms yield comparable performance to the original greedy algorithm, while exhibiting significantly improved time efficiency.

The rapid expansion of real-world networks has imposed higher requirements on the scalability and efficiency of algorithms. In recent years, significant research progress has been made in applying swarm intelligence algorithms to solve the IM problem. Various strategies have been developed to address this challenge using swarm intelligence techniques, such as the Artificial Bee Colony (ABC) [11], Ant Lion Optimizer (ALO) [12], Whale Optimization Algorithm (WOA) [13], Gray Wolf Optimizer (GWO) [14]. A summary of some of the pertinent study findings is given in Table 1. The summary in Table 1 indicate that, despite their effectiveness in identifying influential nodes, swarm intelligence-based approaches often entail high computational complexity when applied to the IM problem under standard propagation models.

## Hybrid-based approaches

In recent years, scholars have integrated the fundamental concepts of network characteristics with swarm intelligence algorithms to enhance efficiency and performance. For example, in the DBA algorithm, if the local search is limited to the Clique structure, the convergence and stability of the algorithm will be improved [23]. Gong *et al.*[24] presented a community-based memetic algorithm to address the IM problem, and the author's experimental results in a real social network showed that its performance was 12.5%,13.2% and 173.5% higher than the Degree Centrality, PageRank and

**Table 1. Summary of swarm intelligence algorithms of solving IM problems.**

| Algorithm | Full Name | Diff Model | Complexity |
|---|---|---|---|
| DSFLA [15] | Discrete shuffled frog-leaping algorithm | IC | $O(mnk\overline{D} + (nm\log(nm) + mk\overline{D})g_{max})$ |
| DBA [16] | Discrete bat algorithm | IC | $O(nk(\log k + \overline{D})g_{max})$ |
| SAEDV [17] | Simulated Annealing Expected Diffusion Value | IC | $O(Tk\overline{D})$ |
| DPSO [18] | Discrete particle swarm optimization Enhanced discrete | IC, WC | $O(k^2 n\overline{D}^2 g_{max}\log k)$ |
| ELDPSO [19] | particle swarm optimization | IC | $O(k^2 n\overline{D}^2 g_{max}\log k)$ |
| DFMO [20] | Discrete Moth-Flame Optimization | IC | $O(n(k\overline{d}d^2 + n)T_{max})$ |
| IM-SSO [21] | IM-Social Spider optimization | LT, IC | $O(n\left(k\overline{D}(N)^2 + n\right) + n\log n)$ |
| ACO-IM [22] | Ant Colony Optimization | IC, WC, LT | – |

Random algorithm, respectively. Taherinia *et al.* [25] introduced the LGFIM algorithm as a solution to the IM problem in large-scale social networks. This approach consists of two stages, the search space is optimized in the first stage using community detection as well, and the second stage uses three heuristic algorithms to adjust the optimal community structure. The increasing scale of real-world networks has driven researchers to prioritize not only the performance but also the computational efficiency of algorithms used to address IM problems.

The aforementioned hybrid algorithms, which combine swarm intelligence optimization algorithms with network centrality, significantly enhance effectiveness compared to single methods. However, in large-scale networks, its time complexity does not decrease but increases.

## Deep learning-driven approaches

With the rapid advancement and success of deep learning across multiple domains, an increasing number of researchers have turned their attention to applying these techniques for solving the IM problem. Wang et al. [26] proposed an end-to-end trained dual-coupled graph neural network algorithm called DGN for the selection of seed nodes in IM problem. Chen et al. [27] introduced the ToupleGDD algorithm, which combines three coupled graph neural networks (GNNs) for network embedding to learn the network nodes, thereby identifying the set of the most influential nodes. Kumar et al. [28] utilized graph embedding and GNN to transform the IM problem into a pseudo-regression problem and proposed a method called SGNN algorithm to solve the problem of maximizing influence in large-scale social networks. Based on the adjacency matrix of the network structure and the convolutional neural network, Yu et al. [29] proposed an effective method RCNN to identify the set of key nodes with the strongest propagation in the networks. Ou et al. [30] integrated the multi-layer structure attributes into the RCNN model, resulting in the Multi-Channel RCNN (M-RCNN) model. The experiments showed that compared with RCNN algorithm, M-RCNN achieved an average accuracy improvement of 9.25%. While deep learning-based methods have shown considerable potential in solving the IM problem, they still face certain limitations in comparison with swarm intelligence-based and other conventional optimization methods. For instance, deep learning-based methods require label for training. However, in real-world networks, such label information is often either unavailable or of low quality. Moreover, most of these approaches neglect the overlapping influence among nodes.

Inspired by the operational principles of convolutional neural networks, this paper proposes a parallel computing-based framework for swarm intelligence optimization algorithm to address the IM problem. The approach effectively combines the strengths of swarm intelligence with the efficiency gains of parallel computing, achieving both high performance and improved computational speed.

## Preliminaries

**Definitions. Definition 1** (Social Network) Give a graph network $G(V, E)$, V denotes a set of individuals and E denotes a set of relationships between individuals. $|V|$ and $|E|$ are the number of nodes and edges in the graph network, respectively.

**Definition 2** (Multithreaded parallel computing) The technique of multithreaded parallel computing is extensively utilized in modern multi-threaded operating systems that run on multi-core processor architectures. It allows for the simultaneous execution of multiple threads on a single machine, where each thread can handle a separate task.

**Definition 3** (N-hop neighbor node set) The N-hop neighbor nodes of a node are those nodes that can be reached from the node through the minimum and exact number of $N$ edges. A set of such nodes with these characteristic forms the N-hop neighbor set of the node.

**Definition 4** (Seed node set) A collection of nodes in the set that can both act as the source of some information and output some information to its direct neighbors is known as the seed node set. In social networks, it can be expressed as $|S| = k, S \in V$.

**Definition 5** (Influence Maximization) For a given seed node set $S$ and a specific diffusion model $P$, the influence spread $P(S)$ is defined as the expected number of influenced nodes after the completion of the influence propagation process. The goal of influence maximization (IM) is to select a seed set $S \in V$ of size K such that $P(S)$ is maximized, i.e., Max $\{P(S) : |s| = k, S \in V\}$.

**Definition 6** (Overlapping Influence) The overlapping influence that exists among nodes in the actual network. Let the influence of node $A$ be denoted by $f(A)$, and that of node $B$ by $f(B)$. Let $f(A \cup B)$ represent the combined influence of nodes $A$ and $B$. In real-world networks, there typically exists $f(A \cup B) \leq f(A) + f(B)$, reflecting the presence of overlapping influence.

## Diffusion model

To effectively analyze and solve the IM problem, it is essential to adopt an appropriate diffusion model that simulate the mechanisms of influence spread across the network. The basic influence propagation models commonly used in social network analysis include the Independent Cascade (IC) model, Weighted Cascade (WC) model, Linear Threshold (LT) model, and Susceptible Infection (SI) model. In each of these models, a node can exist in one of two states: activated (influenced) or inactive (uninfluenced). From the existing research, many investigations on the IM problem based on IC model and WC model have been conducted [15–18]. In these two models, influence propagates from the seed nodes along the time series. For example, only the seed nodes are active at time $T_0$, and these nodes try to activate the inactive state nodes in their one-hop neighborhood with probability $p$, and $P(S)$ nodes are successfully activated. At time $T + 1$, the set of nodes in the active state is $S = S \cup P$, and the inactive state nodes in the one-hop neighborhood of these active state nodes are repeatedly activated in the same way. Until no additional nodes are activated in the network, this activation process keeps going. Let the probability that node v is successfully activated by node u be $p_{u,v}$, then $p_{u,v}$ can be expressed as:

$$p_{u,v} = \begin{cases} [0.01, \ 0.1] & \textit{Constant} & (1) \\ \dfrac{1}{\textit{degree}(v)} & \textit{Weighted Cascade.} & (2) \end{cases}$$

In the IC model, the activation probability $p_{u,v}$ is determined by Eq (1), which is an independent constant. In the WC model, the activation probability $p_{u,v}$ is determined by Eq (2), which is determined by the degree of the node v. In this paper, the IC model is used to simulate influence spread.

## Fitness function

In treating the IM problem as an optimization task, the influence spread is often approximated by a simplified fitness function, which serves as the target for optimization. Various optimization strategies are employed to identify the set of seed nodes that maximizes this function. Jiang *et al.* [17] presented an Expected Diffusion Value (EDV) function that is used to approximate the influence of the seed node set in simulated annealing algorithm. The EDV function approximates the influence of the seed node set based on the one-hop near-neighborhood node set of the seed node set. Gong *et al.* [18] introduced the Local Influence Estimation (LIE) objective function based on two-hop neighbor nodes of seed nodes, which has a good approximate effect in many optimization algorithms [19,23]. The LIE function is shown as:

$$\text{LIE} = \sigma_0(S) + \sigma_1^*(S) + \sigma_2^{\sim}(S)$$

$$= k + \sigma_1^*(S) + \frac{\sigma_1^*(S)}{\left| N_S^{(1)} \backslash S \right|} \sum_{u \in N_S^{(2)} \backslash S} p_u^* d_u^*$$

$$= k + \left( \frac{1}{\left| N_S^{(1)} \backslash S \right|} \sum_{u \in N_S^{(2)} \backslash S} p_u^* d_u^* \right) \sum_{i \in N_S^{(1)} \backslash S} \left( 1 - \prod_{(i,j) \in E, j \in S} \left( 1 - p_{i,j} \right) \right)$$

(3)

where $N_S^{(1)}$ is the sum of node degrees in a node's one-hop neighborhood; and $N_S^{(2)}$ is the sum of node degrees in the two-hop neighborhood of the node. $p_u^*$ represents the probability that one node will activate its neighboring nodes successfully every time. $d_u^*$ is the number of edges between the set of one-hop neighbors and the set of two-hop neighbors of node u. In this work, the LIE function serves as an approximate evaluation of the local influence of a node.

## Basic crow search algorithm

The basic Crow Search Algorithm (CSA) is a bionic intelligence optimization algorithm that is derived from the behavior of crows to hide and find food [31]. The basic CSA models the foraging behavior of a group of N crows in a d-dimensional search space. Each crow is represented by a position vector that denotes its current location, and a memory vector that stores its best-found solution over iterations. The position vector of crow $i$ in the $t$-th generation iteration can be expressed as $X_i^t (i = 1, 2, \ldots N; t = 1, 2, \ldots, t_{max})$, where $t_{max}$ is the maximum iteration. The memory vector of crow $i$ can be expressed as vector $m_i^t$, which is the best location crow $i$ stored its food. The food hiding location of each crow in the flock throughout the iteration is stored in a memory matrix M.

The optimization mechanism of basic CSA is inspired by the greedy behavior of crows. The algorithm simulates the process by which crows follow each other to exploit food sources more effectively, guiding the search toward optimal solutions. At iteration t, let $m_j^t$ denote the food source location associated with crow j. Crow i may choose to follow crow j in an attempt to access this potentially food source, thereby updating its own position $x_i^{t+1}$ based on this observation. In this case, the new location vector $x_i^{t+1}$ is determined by whether crow $j$ perceives the tracking behavior of crow $i$. This behavior of tracking versus anti-tracking can be expressed as:

$$x_i^{t+1} = \begin{cases} x_i^t + r_i * fl_i^t * \left( m_j^t - x_i^t \right), & r_j < AP_j^t \\ a \ random \ positon. & r_j \geq AP_j^t \end{cases}$$

(4)

where $r_i$ is a uniformly distributed random number in the range of 0 and 1, $AP_j^t$ represents the perceived probability of crow j, $fl_i^t$ denotes the flight length of crow $i$ in the current iteration. $AP_j^t$ and $fl_i^t$ are two important parameters that control the search range of the algorithm, where $fl_i^t$ represents the search length of the crow, and its meaning in the basic CSA algorithm is shown in Fig 1(a).

## Proposed method

By discretizing the position and memory vectors of the basic CSA and reconstructing its search mechanism, a Parallel Discrete Crow Search Algorithm (PDCSA) tailored for solving the IM problem is proposed in this paper.

## Discretized coding and optimization rules

The implementation of the PDCSA begins with a preprocessing stage in which all nodes in the network are assigned unique identifiers. This is followed by the discrete encoding of the position and memory vectors. Based on this encoding, the algorithm reconstructs its search and update mechanisms to work within the discretized framework.

**Network node recoding:** Given a network G, let the number of nodes be N, and re-encode each node with a positive integer from 1 to N to ensure the uniqueness of node number in the network. The position vector and memory vector are defined and encoded based on the network node recoding.

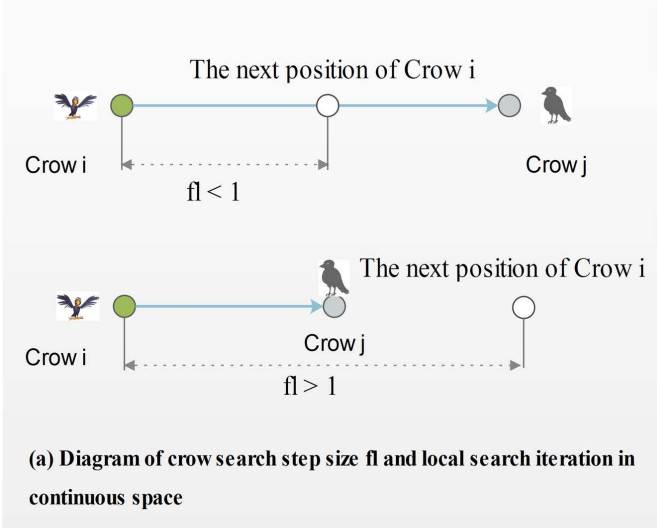

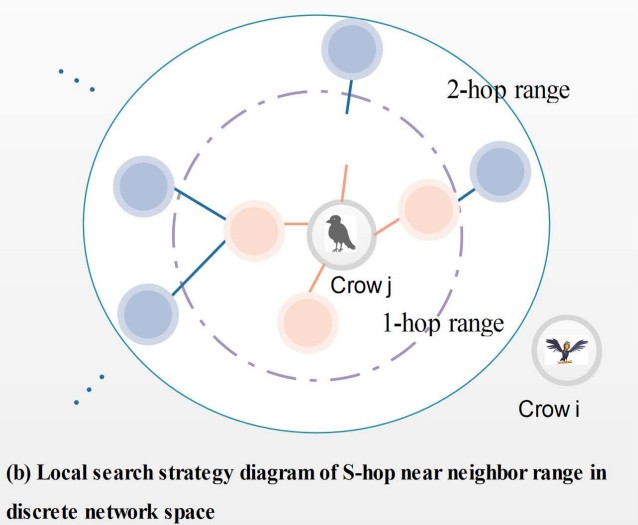

(a) Diagram of crow search step size fl and local search iteration in continuous space

(b) Local search strategy diagram of S-hop near neighbor range in discrete network space

**Fig 1. Local search diagram based on crow tracking and anti-tracking behavior.**

**Position vector encoding:** With K designated as the number of seed nodes, the position and memory vectors each consist of K nodes that represent a current or previously found optimal solution. The position vector of crow i can be expressed as $x_i = (node_1, node_2, \cdots, node_k)$ where $node_{1\ldots k}$ represents the node code of network in the PDCSA. The position vector of crows in generation t is expressed as:

$$Crow_t = [x_1, x_2, \cdots, x_M]^T \tag{5}$$

where $M$ represents crows in the crow population.

**Memory vector encoding:** Memory vector is employed to maintain the optimal solution throughout the search process. If $t$ is the current iteration number, the memory vector saves the food hiding position of the crow after the search in the previous t-1 generation, which can be expressed as:

$$Memory_{t-1} = [m_1, m_2, \ldots, m_M]^T \tag{6}$$

where $m_i = (node_1, node_2, \ldots, node_k)$ represents the best location to hide food for crow *i* in the *t-1* generation.

The encoding of these three types of vectors and their relationships are shown in Fig 2(b). As the optimization proceeds, the position and memory vectors of each crow are updated by replacing their constituent nodes to improve solution quality.

**Optimization rules reconstruction:** Based on the above encoding rules, the discretization optimization process of PDCSA is constructed as:

$$X_i^{t+1} = \begin{cases} X_i^t \oplus R(r_i, S) \bullet \left(m_j^t \cap x_i^t\right) & r_j \geq AP_j^t \tag{7} \\ Random\ Selection(N) & otherwise. \tag{8} \end{cases}$$

where the symbol ∩ represents a logical intersection operation that aims to determine if two vectors contain identical nodes. If the node $m_j^h$ in the memory vector $m_j$ exists in the position vector $x_i$ this operation returns 0, otherwise, returns

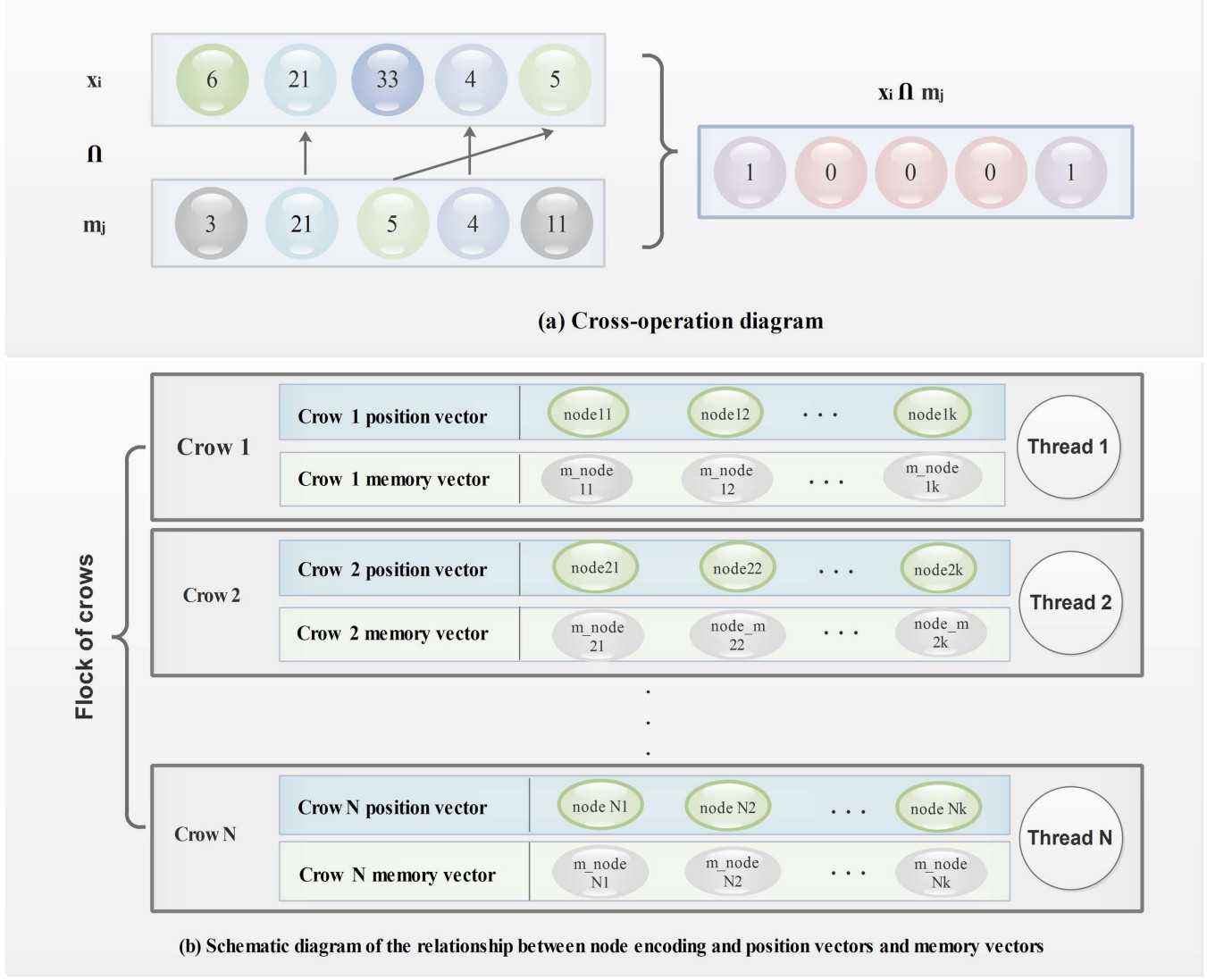

(a) Cross-operation diagram

(b) Schematic diagram of the relationship between node encoding and position vectors and memory vectors

**Fig 2. Schematic diagram of PDCSA coding and optimization.**

1. For example, if the memory vector of crow j is $m_j(1, 21, 5, 4, 11)$, and the position vector of crow $i$ $x_i(3, 21, 2, 5, 17)$, the crossover operation is shown in Fig 2 $R(r_i, S)$ is a local search mechanism limited by parameters $r_i$ and $S$ where $r_i$ represents the probability of randomly selecting the nearest neighbor node of a node in the current position vector. $S$ represents the S-hop range of the nearest neighbor of a node in the current position vector and its meaning is shown in Fig 1(b) The operator $\oplus$ in Eq (7) is used for the replacement operation. If the result value of the corresponding position after the $\cap$ operation is 1, the node is updated and replaced by the $R(\cdot)$ operation.

## Framework of PDCSA

Based on the basic CSA algorithm framework, along with the aforementioned discrete encoding strategy and optimization rules, the PDCSA algorithm is structured into five main steps:

**Step 1 Initialization**. Set the control parameters: crow population size $M$, seed node set size $K$, maximum iterations $t_{max}$, local search range $s$, perception probability $AP$. Initialize the position and memory vectors of each crow with K nodes.

**Step 2 Calculation of the LIE value**. Using Eq (3), the algorithm evaluates the LIE value for each node present in the position vector of every crow. This step supports subsequent optimization decisions by identifying nodes with higher local influence potential.

**Step 3 Generation of New Position Vectors**. The new position vector of Crow $i$ is generated based on either Eq (7) or Eq (8). Subsequently, the LIE values of all nodes in the new vector are evaluated, and the most favorable LIE value is selected to form the updated position vector.

**Step 4 Update of the Memory Vector**. According to Eq (9), the memory vector of each crow is conditionally updated based on the comparison between the LIE values of the current and previous position vectors. Specifically, if the influence potential of the new position vector exceeds that of the former, the memory vector is replaced with the new one; otherwise, it remains unaltered.

$$m_i^{t+1} = \begin{cases} x_i^{t+1}. & \textit{If LIE}\left(x_i^{t+1}\right) \textit{ is better than LIE}(m_i^t) \\ m_i^t . & \textit{otherwise}. \end{cases} \tag{9}$$

**Step 5 Termination Check**. The algorithm proceeds to check whether the current iteration count has reached $t_{max}$. If not, the optimization loop (Steps3–4) continues. Once the maximum iteration limit is attained, the algorithm identifies the globally best-performing seed node set by selecting the memory vector with the highest LIE value as the final solution.

The flowchart depicting the overall structure and execution steps of the proposed PDCSA algorithm is shown in Fig 3.

## The implementation of PDCSA

A detailed description of the PDCSA algorithm for solving IM problems, based on the five steps outlined above, is provided in Algorithm 1. It is worth mentioning that when initializing the position vector of the crows, we select $M \times K$ nodes with the highest degrees from the network to form the initial seed node set. This strategy helps accelerate the convergence speed by starting the search from more influential candidate solutions. Algorithm 1 consists of two key functions: LocalSearch(), which updates the position vectors based on Eq (7) to enhance solution accuracy, and RandomExploration(), which employs Eq (8) to diversify the search process and escape local optima.

The PDCSA algorithm is designed with a decentralized structure: each crow operates with its own position and memory vectors, as depicted in Fig 2(b). The optimization proceeds by iterating over $M$ loops, where each loop involves only pairwise interactions between crows. As these operations do not require shared computation or synchronization among crows, the algorithm naturally lends itself to parallel execution.

In this study, OpenMP is used to conduct parallel computing experiments. In the Algorithm 1, the statement #*pragma omp parallel* for specifies the parallel region and instructs the compiler that the following for loop should be executed in parallel. meaning that the iterations of the loop are distributed across multiple threads for concurrent execution. In the experimental environment with multicore multithreaded processors, when performing parallel computing in C++, only Step 3 and Step 4 of PDCSA framework that require parallel computation need to be marked with the #*pragma omp parallel* directive at the beginning of their respective code. After that, the compiler will automatically use multi-thread for independent iterative operations. In the PDCSA algorithm framework, after Step 3 and Step 4, use #*pragma omp* barrier directive to wait for all threads to complete their tasks before proceeding with the selection of the local optimum, thereby completing one iteration.

Start

**Step 1: Initialize the parameters and some vectors**

a. Flock of crows size M
b. Seed node set size K
c. Maximum number of iterations $t_{max}$
d. Local search node neighbor range S
e. Perception probability AP
f. Position vectors of crows
h. Memory vector of each crow

**Step 2: Calculate the LIE value**

Calculate the LIE value of the node in the position vector of each crow

**Step 3: Create new posion vector**

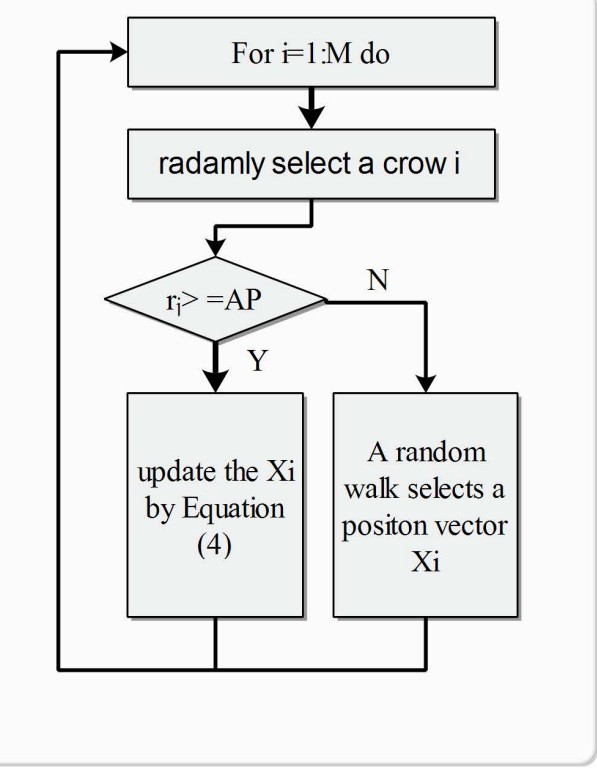

For i=1:M do

radamly select a crow i

$r_j >= AP$

N

Y

update the Xi by Equation (4)

A random walk selects a positon vector Xi

**Step 4: Update memory vector**

The memory vector is updated according to Equation (9)

**Step 5: Check the termination condition**

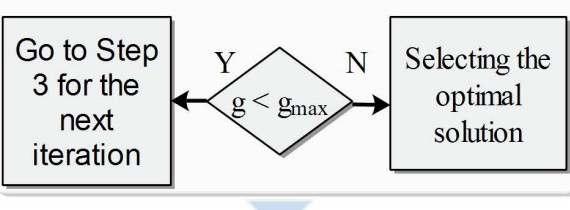

Go to Step 3 for the next iteration

Y

$g < g_{max}$

N

Selecting the optimal solution

End

**Fig 3. The flowchart of the PDCSA algorithm.**

## Algorithm 1. Framework of PDCSA algorithm based on parallel computing

```
Input: G = (V, E), seed node set size K, maximum number of iterations g_max, Crow population size N
Output: The best seed set S_node
1:  Initialize iterator g=0
2:    Define the perception probability AP
3:  Position vector X←Select K*N nodes
4:  Memory vector M←Position vector X
5:  WHILE g<g_max {
6:    #pragma omp parallel for
7:    FOR i=1:M (all crows of the flock) {
8:        A crow j is randomly selected as the tracked object
9:        A random number r_j is generated
10:        if r_j≥AP
11:            Xi←LocalSearch(V): According to Eq. (7)
12:        else
13:            Xi ← RandomExploration(V): This corresponds to Eq. (8)
14:        endif
15:    }
16:  #pragma omp barrier
17:    Evaluate the new position vector of the crows
18:    Update the memory vector of crows according to Eq (9)
19:  }
20:  S_node←Select the Max LIE value from the memory vectors
```

### Local and global search mechanisms

**Local search strategy.** In the PDCSA algorithm, the execution of a local search is determined by the perception probability *AP*. Specifically, a local search is conducted when the random probability *p* satisfies *p > AP*. Since the position vector is composed of *K* nodes, the particular node $x_i$ selected for local search within the position vector is determined through the intersection operation "∩" between the memory vector *m* and the position vector *x*, as formally defined in Eq (10):

$$i = (m_j \cap x_i), i \in \{0, 1\}^k \tag{10}$$

A local search operation is applied exclusively to node $x_i$ under the condition that i = 1. The local search procedure is mathematically formulated in Eq (11):

$$x_{i\_new} = argMax \{ LIE(x_i),\ LIE(r_i \cdot NodeSet_s) \} \tag{11}$$

where $NodeSet_s$ denotes the set of S-hop neighboring nodes of node $x_i$, and the random number $r_i$ is used to select a node within this neighborhood.

Based on the aforementioned formulation of search strategy, the local search rule is defined as Eq (7). Specifically, the algorithm computes the intersection $m_j^t \cap x_i^t$ between the current position vector $x_i^t$ of crow I and the memory vector $m_j^t$ of the tracked crow j. This intersection operation identifies the different nodes between the two vectors. A decision to perform a local search around node in the position vector is determined by the outcome of the intersection result. A value of 1 at a specific position indicates that the neighborhood of this node requires a local search. Conversely, if the value is 0, the node remains unaltered.

## Algorithm 2. Local search strategy based on neighbor node domain

```
Input: x_i^t, m_j^t, S
Output: x_i_new
1:  V_node ← m_j^t ∩ x_i^t
2:  For(i=1:N)do {
3:     IF V_node(i)==1 THEN
4:         NBNodeSET_s ←CalculateNBNode(V_node(i),S)
5:         LIEValue = LIECalculate(x_i^t,NBNodeSET_s)
6:         x_i_new←MaxLIENode(LIEValue)
7:   END IF
8: }
```

Given that crows' positions are represented as set of nodes in the network, their flight length is not Euclidean but rather a shit among connected nodes. Therefore, the local search scope is naturally constrained to the nodes included in the current position vector, guiding the search toward locally optimal solution. Fig 1(b) shows the local search mechanism based on the S-hop neighbor node set. In the S-hop local search mechanism, the value of $S$ is natural number, i.e., $S \leq N(N = 1, 2, \ldots, N)$. The parameter S determines the depth of the local search within the network. For S=1, the search is confined to immediate neighbors of the node for which the intersection operation result in the position vector is 1, as shown by the dotted circle in Fig 1(b). For S=2, the search includes nodes at two hops domain, indicated by the solid circle. Higher values of S enlarge the search scope to encompass more flight distant neighbors. The S-hop-based local search mechanism is described in Algorithm 2.

In Algorithm 2, Function CalculateNBNode() computes the S-hop neighbor node domain of node $V_{node}$ (i) and returns its S-hop neighbor node set, and its pseudo-code is described in Algorithm 3. The function LIECalculate() calculates the LIE value of each node in the $x_i$ position vector replaced by the node of NBNodeSET_s set. The function MaxLIENode() selects the node that has the best LIE value from the NBNodeSET_s set. The function MaxLIENode() selects the node that has the best LIE value from the NBNodeSET_s node-set returned by the function LIECalculate(), replaces the corresponding node in the position vector $x_i$, and there are no duplicate nodes in the position vector $x_{i\_new}$ after the replacement operation.

## Algorithm 3. CalculateNBNod(): Search for S-hop neighbor node-set

```
Input: S, V_node(i)
Output: The S-hop area node set: Nodeset
1:  Neighbors←DirectNeibors(V_node(i))
2:  Nodeset←Neighbors
3:  Neighbor←∅
4:  Step=1
5:  WHILE(Step <S) {
6:    FOR node∈Neighbors do
7:        Nodeset← Nodeset ∪ DirectNeibors(node)
8:        Neighbor←Neighbor ∪ DirectNeibors(node)
9:    END FOR
10:   Neighbors←Neighbor
11:   Step←Step +1
12: }
13: Nodeset←RemoveDuplicates(Nodeset)
```

In Algorithm 3, the first-order neighbors of node $v_{node}(i)$ are stored into node set Neighbors, and then the node set Neighbors is traversed to find out the neighbor nodes of its S-hop and merged into the node set NodeSet. After the S-hop neighbor nodes of each node are traversed, duplicate nodes in NodeSet are removed to ensure that there are no duplicate S-hop neighbor nodes.

## Global exploration strategy

Global exploration is carried out when the randomly generated probability value is less than the perceived probability $AP$. To ensure the global exploration capability of the algorithm, the DCSA algorithm adopts random numbers based on uniform distribution to realize random walks. Given that the number of network nodes is $V$, the number of seed nodes is $K$, and the size of the crow swarm is $N$. The global search space $L$ is mathematically represented as the collection of all possible subsets of size $K$ selected from a set of $(V - N * K)$ candidate nodes. Let $Z = V - N \times K$, the solution space is defined as in Eq (12), and its size, denoted by $|L|$, is specified in Eq (13).

$$L = \{U \subseteq Z : |U| = K\} \tag{12}$$

$$|L| = \binom{Z}{K} \tag{13}$$

Given a perception probability $AP$, the probability of triggering a global search in any iteration is $1 - AP$. Assuming the search process follows a geometric distribution, the expected number of iterations t can be derived as:

$$t = \frac{|L|}{1 - AP} \tag{14}$$

It is evident that the random walk-based global search asymptotically converges to the global optimal solution over the course of the iterative process. The random walk strategy is described in Algorithm 4.

## Algorithm 4. Global exploration based on random walk strategy

```
Input: seed node set size K
Output: Xᵢⁿᵉʷ
1:  index← 1
2:  Xᵢⁿᵉʷ←∅
3:  WHILE index<= K DO {
4:     Node_temp← RandomSelect(V, index)
5:     IF Node_temp not in Xᵢⁿᵉʷ THEN
6:         Xᵢⁿᵉʷ←Xᵢⁿᵉʷ ∪ Node_temp
7:     END IF
8:     index = index+1
9:  }
```

## Computational complexity

The performance of an optimization algorithm in solving IM problems is typically assessed based on two aspects: its computational complexity, indicating runtime efficiency, and its effectiveness, which evaluates how well the algorithm identifies influential seed nodes. In this section, we make a theoretical analysis of the computational complexity of the PDCSA algorithm. Given the number of network nodes be $N$, the maximum number of iterations $g_{max}$, the seed node set size of $K$ and the average degree of the network be $\overline{D}$. The computational complexity of the algorithm is as follows without considering the parallelization operation. In the local search of the S-hop, the time complexity of the search nodes' S-hop neighbor operation is $O(K \times \overline{D}^S)$, the update Xi operation is $O(K)$. The time complexity of global search operation based on random walk is $O(N)$, updating memory vectors is $O(N \times K)$, calculating LIE value is $O(K \times \overline{D})$. Therefore, the time complexity of the above calculation is $O\left(K \times \overline{D}^S + K \times \overline{D} + K + N \times K\right)$. According to the operation rule of the symbol $O$, the computational complexity of the PDCSA is $O\left(K \cdot g_{max} \cdot (\overline{D}^S + N)\right)$.

Based on the above analysis of computational complexity, it can be observed from Table 1 that the computational complexity of the PDCSA algorithm is comparable to, and in some cases slightly better than, that of state-of-the-art swarm intelligence algorithms. A comparison shows that the computational complexity of the PDCSA algorithm is comparable to that of DPSO and ELDPSO, but inferior to that of DBA and IM-SSO algorithms. The aforementioned comparison is derived from theoretical analysis. It is noteworthy that the PDCSA algorithm inherently amenable to parallel implementation on multi-core processors. Consequently, when the speedup from parallelization is considered, its actual efficiency has the potential to surpass that of state-of-the-art swarm intelligence optimization algorithms.

## Experiments

In this section, a series of experiments are conducted to evaluate the performance of the PDCSA algorithm. First, we determine the optimal parameter settings for PDCSA across six experimental networks. Next, the LIE values are compared with those of other swarm intelligence-based algorithms. Finally, the performance and the time efficiency are compared with the state-of-the-art algorithms.

The relevant algorithm is implemented in C++, and the experiments are carried out on a PC platform equipped with an Intel(R) Cores (TM) i7-8700 CPU running at 4.6GHz with 32GB of RAM. The Independent Cascade (IC) propagation model is used for influence spread evaluation, and the maximum size of the seed node set is set to 50 in all experiments.

## Datasets and baseline algorithms

To evaluate the effectiveness of the proposed PDCSA algorithm, six large-scale real-world networks are selected for experiment. Table 2 summarizes the characteristics of the six experimental networks used in this study. Among them, SynRand is a synthetically generated network with a Gaussian degree distribution, consisting of 14,991 nodes and 56,152 edges. PGP is a social network based on the Pretty Good Privacy (PGP) encryption algorithm, modeling trust relationships among 10,680 individuals who exchange encrypted messages. CondMat is a collaborative network for published articles in the field of physics. Slashdot is a social network constructed from user interactions on the technology news website. All networks are obtained from the Stanford Network Analysis Project (SNAP) repository.

In the evaluation of LIE values, algorithms DPSO [18], DBA [16], AMPDE [36], DPSO_NDC [37] and Clique_DBA [23] are chosen as comparative baselines. A Distinguishing feature of these algorithms is their shared use of the LIE function as the optimization target in the optimization process. The other three most advanced algorithms serve as baseline algorithms to compare the performance of PDCSA, which are CELF [9], Greedy algorithm [5] and CLDE [38]. A comparative experiment is carried out under the propagation probability $p = 0.01$ and $p = 0.05$.

- CELF (Cost-Effective Lazy Forward) is an improved greedy algorithm based on strategy lazy forward-selection, which significantly improves efficiency while maintaining high performance.

**Table 2. Network characteristics of six experimental real-world networks.**

| Networks | Num-of-Nodes | Num-of-Edges | Ave-Degree | Ave-Path | Ave-Cluster | Density |
|---|---|---|---|---|---|---|
| Email [32] | 1133 | 5452 | 9.624 | 3.606 | 0.254 | 0.009 |
| Slashdot [32] | 77360 | 905468 | 23.41 | 4.025 | 0.085 | 0.009 |
| Artist [33] | 50515 | 819306 | 16.22 | 5.945 | 0.601 | 0.001 |
| PGP [34] | 10680 | 24316 | 4.55 | 7.486 | 0.44 | 0.001 |
| CondMat [35] | 23133 | 186936 | 16.16 | 5.353 | 0.706 | 0.001 |
| SynRand | 14991 | 56152 | 7.49 | 5.006 | 0.001 | 0 |

- DBA (Discrete Bat Algorithm) is a discrete Bat Algorithm based on the network structure, which mainly simulates the foraging behavior of bat swarms to achieve optimization. Experimental results show that the DBA algorithm has advantages in both performance and efficiency in large-scale networks.

- DPSO (Discrete Particle Swarm Optimization) is a swarm intelligent optimization algorithm based on recoding and reconstructing the evolutionary rules of the basic Particle Swarm Optimization algorithm to maximize the influence of social networks.

- The Greedy algorithm is based on the greedy strategy, which is based on the principle of selecting a node with the most significant objective function value from all nodes.

- CLDE (Competitive Learning-driven Differential Evolution) employs a competitive mechanism in which individuals are randomly paired within the population, and each pair competes based on their fitness values to select superior solutions.

- AMPDE (Adaptive Multiple Probabilistic Differential Evolution algorithm) incorporates an adaptive local search mechanism, designed to enhance the search for the optimal solution through the utilization of structural hole nodes and their neighborhoods.

## Parameter configuration

The PDCSA algorithm introduces two primary control parameters when addressing the IM problem in social networks: the nearest neighbor domain S and the awareness probability AP. We determine their optimal values through systematic experimentation on a representative network structure, ensuring balanced exploration and exploitation during optimization.

A systematic grid search was performed using four representative real-world networks, ConMat, SynRand, PGP and Email, to identify the most effective parameter configurations for the PDCSA algorithm. First, the four experimental networks exhibit significant differences in scale. Second, from the perspective of network structure, they include both networks with Gaussian degree distribution (e.g., SynRand) and those with power-law degree distribution (e.g., Cond-Mat, PGP). In the experiment, the crow population size $N$ was set to 30, and the number of seed nodes $K$ was fixed at 30. The awareness probability AP was varied incrementally from 0.1 to 0.9, and the nearest neighbor domain S was expanded from 1 to 5 hops. For each combination of parameter settings, 50 independent runs were conducted to ensure statistical reliability. The average influence spread across these 50 runs was then computed and used to generate the three-dimensional bars shown in Figs 4 and 5. Fig 4 shows the statistical diagram of the LIE value when the propagation probability p = 0.01. As shown in the Fig 4, the optimal LIE values cross the four networks are observed in the parameter region around AP = 0.6 and S=3. As shown in Fig 5, depicting the distribution of LIE values at a propagation probability of p = 0.05, the optimal parameter configuration corresponds well with the results observed in Fig 4. This consistency suggests that the optimal parameter setting is relatively insensitive to variations in propagation probability.

In the PDCSA algorithm, the number of iterations $g_{max}$ is directly related to the algorithm's convergence. Under the previously determined optimal parameter settings (AP=0.6, S=3, K=30), we conducted an exhaustive search to track the evolution process across the four experimental networks under the propagation probability of p=0.01. These observations are graphically presented in Fig 6. It can be observed that the global optimal solution is reached within 200 iterations. Therefore, in the PDCSA algorithm, the maximum number of generations $g_{max}$ is set to 200.

## Comparison of LIE

To evaluate the effectiveness of the PDCSA algorithm, comparative experiments based on the LIE values were conducted across six real-world networks. The experiments were performed under varying sizes of the seed node set K, with values set to 5,10,15,20,25,30,40, and 50, under the propagation probability of p = 0.01. As baseline algorithm, four swarm intelligence algorithms, DPSO, DBA, AMPDE, and CLDE, as well as two improved variants, DPSO_NDC and Clique_DBA,

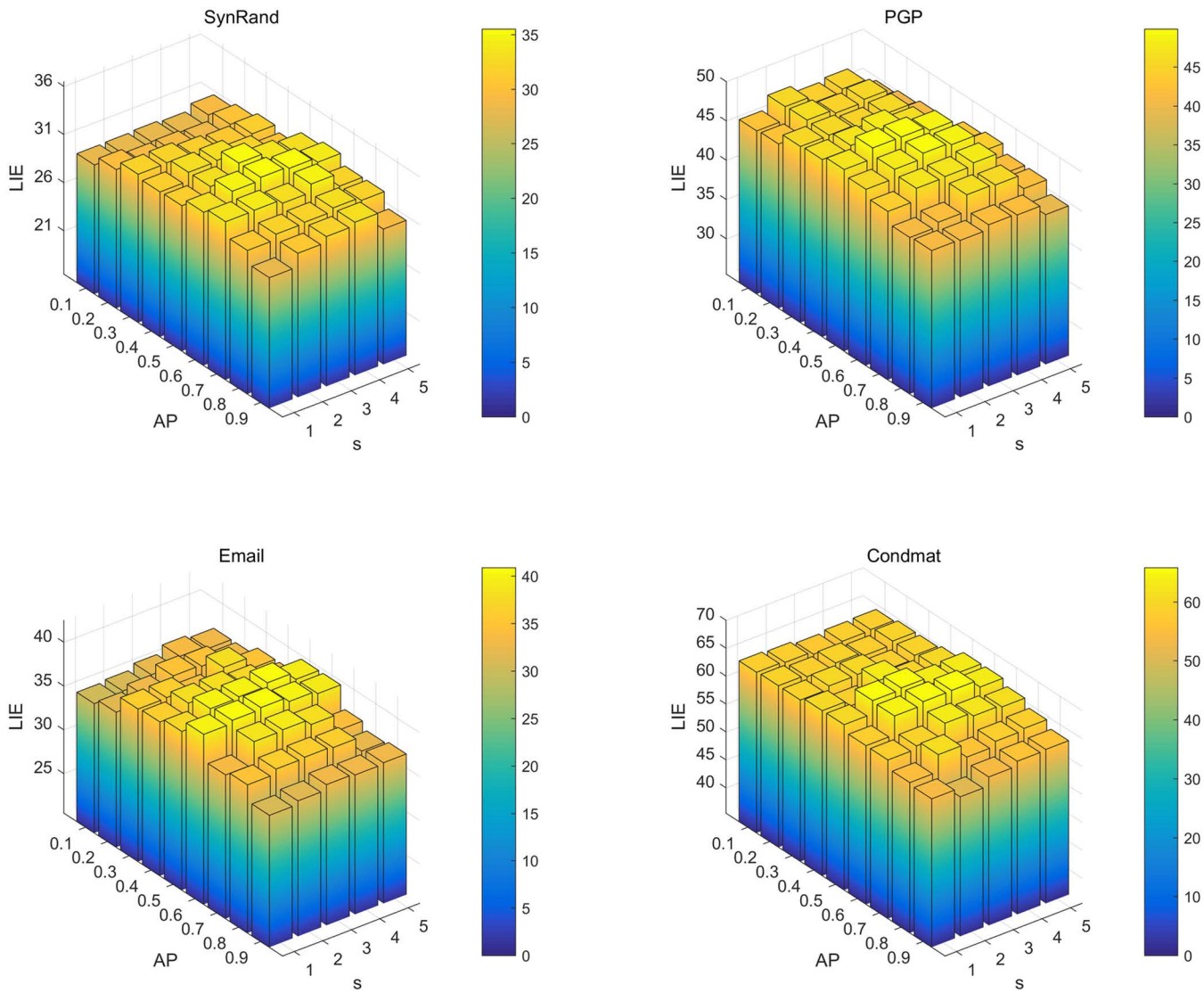

**Fig 4. Experimental statistical graph of LIE values of the nearest neighbor range S and perception probability AP at K = 30 and p = 0.01.**

were selected for comparison. The algorithms involved in the experiment were successfully run 10 times in each of the six real-world networks, and the average value of LIE in the 10 experiments was evaluated. Fig 7 presents the LIE value evolution curves of the compared algorithms under a propagation probability of p = 0.01.

Fig 7 indicates that, the PDCSA achieves LIE values on par with AMPDE, but superior to DPSO and DBA in large-scale networks, as illustrated in Figs 7(a), 7(e) and Fig 7(f). Furthermore, in small and medium-sized networks, its effectiveness are also outperforming the other swarm intelligence algorithms, as shown in Figs 7(b) and Fig 7(c).

## Performance comparison

To evaluate the performance of the proposed PDCSA algorithm, we conducted comparative experiments using the six state-of-the-art algorithms introduced in Section 5.1. The IC model was employed to assess influence diffusion,

**Fig 5. Experimental statistical graph of LIE values of the nearest neighbor range S and perception probability AP at K = 30 and p = 0.05.**

and Monte Carlo (MC) simulation was used to perform 1000 independent propagation runs on the identified seed node sets. The average influence spread across these simulations was taken as the performance metric, providing a reliable estimate of each algorithm's effectiveness under the IC model. Under the conditions that the propagation probability p is set to 0.01, and with seed node set sizes ranging from 5 to 50 (i.e., K = 5,10,15,20,25,30,40, 50), the influence spread was evaluated using MC simulation. The resulting average propagation range curves are presented in Fig 8.

Figs 8(a) and 8(d) show that, at p = 0.01, PDCSA achieves influence spread levels nearly on par with the CELF and Greedy algorithms in the large-scale networks CondMat and Slashdot. This indicates that PDCSA can effectively approximate the performance of traditional greedy methods while maintaining the efficiency and scalability benefits of swarm intelligence-based metaheuristics.

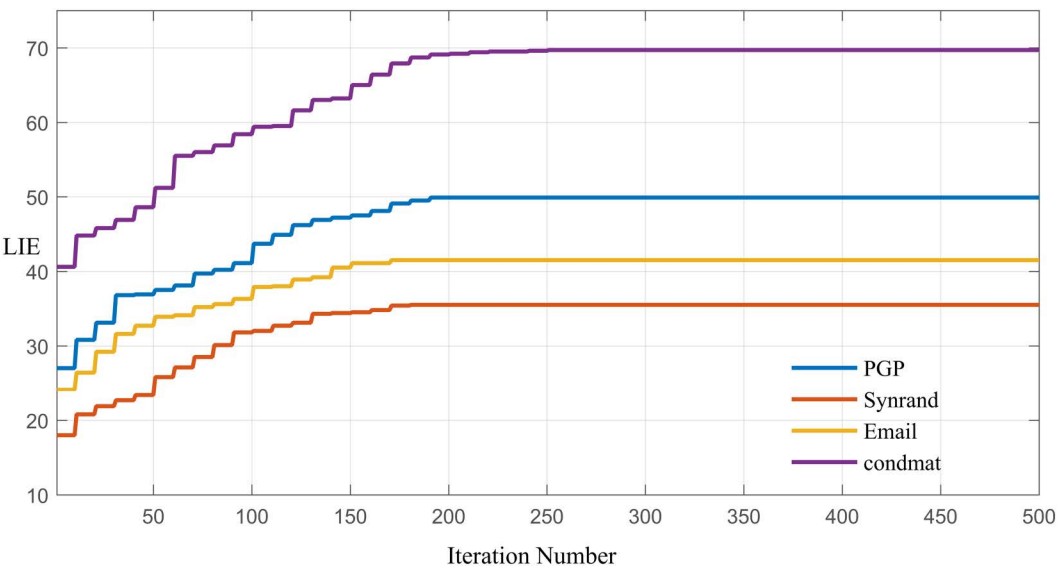

**Fig 6. Evolution of the LIE value over 500 iterations across the four experimental networks.**

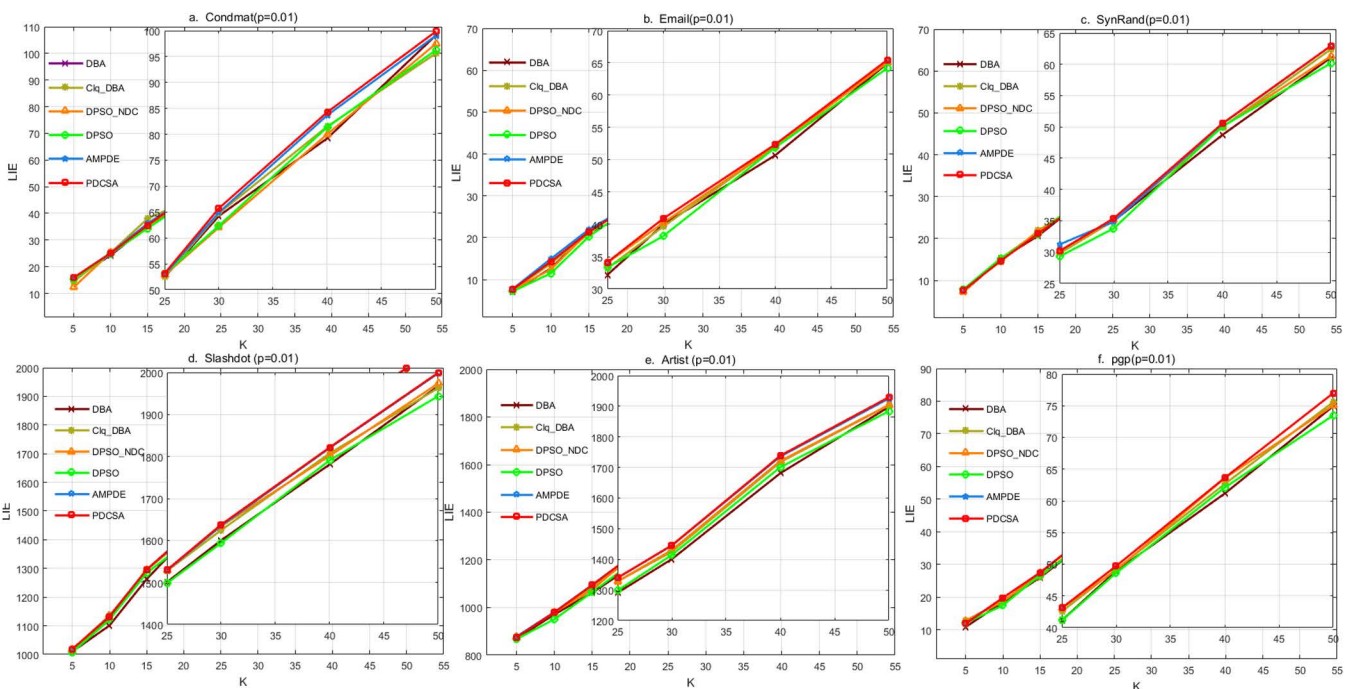

**Fig 7. LIE value carves of different algorithms when propagation probability p = 0.01.**

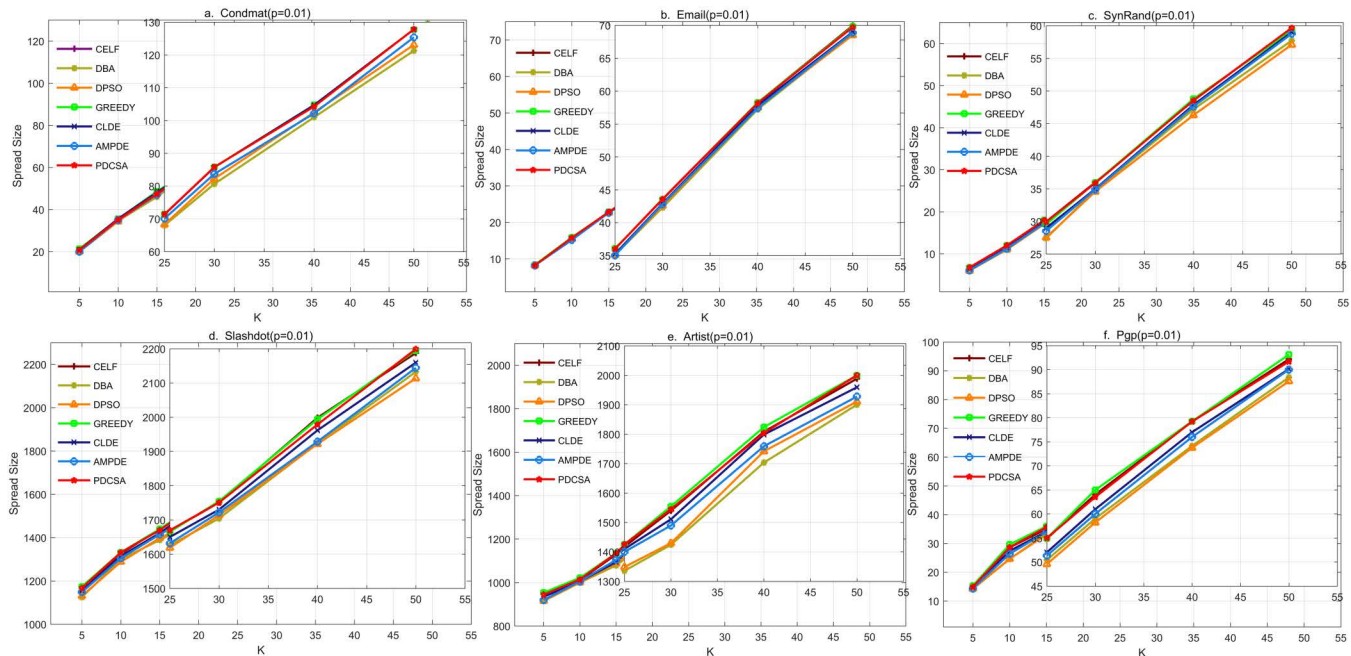

**Fig 8. Line chart of influence propagation results of different algorithms when propagation probability p = 0.01.**

## Running time comparison

To validate the computational efficiency of the PDCSA algorithm, we conducted runtime comparisons with other leading algorithms under identical experimental settings (K = 30, p = 0.01). The average processing times across all six experiment networks were recorded and visualized in Fig 9.

As shown in Fig 9, the general Greedy and CELF algorithms exhibit the highest computational cost across all six experimental networks. Among the remaining four metaheuristic algorithms, the execution times are relatively comparable; however, the PDCSA algorithm demonstrates the highest time efficiency, which can be attributed to its parallel computing capabilities. One of the most distinctive advantages of the PDCSA algorithm lies in the independence of its iterative evolution stages, which enables efficient parallel execution across multi-core architectures. As demonstrated in the runtime comparisons, PDCSA achieves the same or better solution quality as leading algorithms, but with significantly reduced computational time, making it particularly well-suited for large-scale IM tasks. The larger the network scale, the more nitid the time efficiency.

## Conclusions

As social networks grow exponentially in size, traditional algorithms face scalability issues that limit their applicability. One of the most pressing concerns is how to enhance algorithmic efficiency without compromising the quality of results. This challenge motivates the development of novel approaches capable of handling large and complex network structures efficiently. This study introduces a novel metaheuristic approach called Parallel Discrete Crow Search Algorithm (PDCSA), tailored for solving the IM problem in social networks. By integrating parallel computing techniques into its evolution process, PDCSA significantly improves computational efficiency without compromising the effectiveness of the identified seed node set. Table 3 provides a summary of the experimental results for the PDCSA algorithm in comparison with advanced

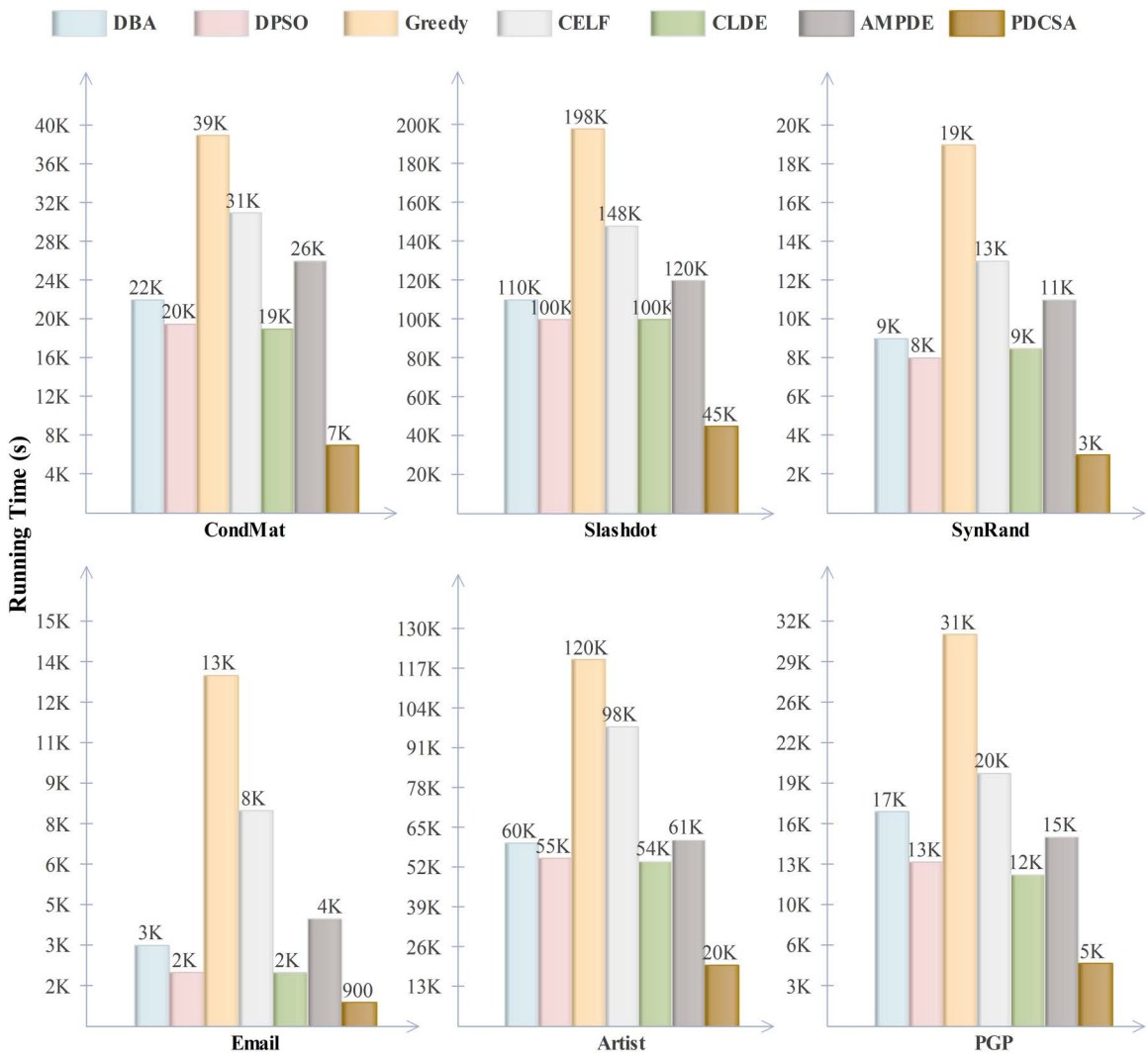

**Fig 9. Statistical graph of processing time of each algorithm in six experimental networks when K = 30 and p = 0.01.**

algorithms across four networks (CondMat, SynRand, Slashdot and Email), under the configuration parameters K = 30 and p = 0.01. As can be observed from the data presented in Table 3, while maintaining solution quality on par with greedy algorithms such as CELF, the PDCSA algorithm offers superior time efficiency compared to other advanced IM methods. This combination of effectiveness and efficiency makes it particularly suitable for real-world applications involving hyper-scale networks.

To efficiently address the problem of influence maximization in large-scale network structure, several important research directions warrant further investigation. First, the application of deep learning techniques to influence maximization in ultra-large-scale networks presents a promising avenue for developing more efficient and scalable algorithms. Exploring such data-driven approaches could lead to significant improvements in both solution quality and computational efficiency. Second, an essential prerequisite for any effective optimization algorithm is the accurate and efficient assessment of node local influence. Therefore, investigating novel methods for evaluating local influence, particularly those that balance computational cost with performance, is a crucial direction for future work.

**Table 3. Experimental results of each algorithm on network CondMat, SynRand, Slashdot and Email under the parameter setting K=30 and p=0.01.**

| Network | Algorithm | Runtime(m) | LIE | Influence |
|---|---|---|---|---|
| CondMat | DBA | 366.7 | 64.2 | 80.7 |
| | DPSO | 333.4 | 62.3 | 82.3 |
| | CELF | 516.7 | – | 85.7 |
| | CLDE | 316.7 | – | 84.9 |
| | PDCSA | 116.7 | 65.7 | 85.6 |
| SynRand | DBA | 150 | 34.9 | 34.8 |
| | DPSO | 133.3 | 33.8 | 34.6 |
| | CELF | 216.7 | – | 36.0 |
| | CLDE | 133.4 | – | 35.0 |
| | PDCSA | 50 | 35.4 | 36.1 |
| Slashdot | DBA | 1833.4 | 1600.6 | 1704.2 |
| | DPSO | 1666.7 | 1594.1 | 1713.7 |
| | CELF | 2166.7 | – | 1750.1 |
| | CLDE | 1668.3 | – | 1730.0 |
| | PDCSA | 781 | 1821.7 | 1750.9 |
| Email | DBA | 50 | 39.9 | 42.3 |
| | DPSO | 33.3 | 38.2 | 42.6 |
| | CELF | 133.3 | – | 43.6 |
| | CLDE | 30.3 | – | 42.9 |
| | PDCSA | 15 | 38.6 | 43.4 |

## Author contributions

**Conceptualization:** Lihong Han.

**Investigation:** Kan Yang.

**Methodology:** Lihong Han.

**Software:** Kan Yang, Yang Ming.

**Validation:** Kan Yang, Jianxin Tang.

**Visualization:** Yang Ming.

**Writing – original draft:** Lihong Han.

**Writing – review & editing:** Lihong Han, Jianxin Tang.

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
