## [Decision Letter · Decision Letter 0]

4 May 2025

PONE-D-24-54198PDCSA: A parallel discrete crow search algorithm for influence maximization in social networksPLOS ONE

Dear Dr. Han,

Thank you for submitting your manuscript to PLOS ONE. After careful consideration, we feel that it has merit but does not fully meet PLOS ONE’s publication criteria as it currently stands. Therefore, we invite you to submit a revised version of the manuscript that addresses the points raised during the review process.

We look forward to receiving your revised manuscript.

Kind regards,

Kuldeep Singh

Academic Editor

PLOS ONE

Journal Requirements:

“This work was partially supported by the Gansu Sci&Tech Program under Grant No. 22JR11RA134, Gansu Provincial Fund for Distinguished Young Scholars under Grant No. 23JRRA766, National Social Science Fund of China under Grant No. 21BTJ042,

Financial Statistics Research Integration Team of Lanzhou University of Finance and Economics under Grant No.XKKYRHTD202304.”

4. We note that your Data Availability Statement is currently as follows: “All relevant data are within the manuscript and its Supporting Information files.”

Please confirm at this time whether or not your submission contains all raw data required to replicate the results of your study. Authors must share the “minimal data set” for their submission. PLOS defines the minimal data set to consist of the data required to replicate all study findings reported in the article, as well as related metadata and methods (https://journals.plos.org/plosone/s/data-availability#loc-minimal-data-set-definition ).

If your submission does not contain these data, please either upload them as Supporting Information files or deposit them to a stable, public repository and provide us with the relevant URLs, DOIs, or accession numbers. For a list of recommended repositories, please see https://journals.plos.org/plosone/s/recommended-repositories .

Reviewers' comments:

Reviewer's Responses to Questions

**Comments to the Author**

1. Is the manuscript technically sound, and do the data support the conclusions?

Reviewer #1: Partly

Reviewer #2: Yes

2. Has the statistical analysis been performed appropriately and rigorously? 

Reviewer #1: N/A

Reviewer #2: Yes

3. Have the authors made all data underlying the findings in their manuscript fully available?

Reviewer #1: Yes

Reviewer #2: Yes

4. Is the manuscript presented in an intelligible fashion and written in standard English?

Reviewer #1: No

Reviewer #2: Yes

5. Review Comments to the Author

Reviewer #1: The paper proposes a new approach to identifying influential nodes within a network, and demonstrates the advantages of the algorithm compared to several existing methods.

Overall, I find this to be a promising and interesting study. However, I have two major concerns:

1. The evaluation of the proposed algorithm relies solely on LIE, an objective function introduced by Gong et al. (2016) in Information Sciences (“Influence Maximization in Social Networks Based on Discrete Particle Swarm Optimization”). In other words, the paper uses a performance measure designed from a swarm intelligence optimization algorithm, to show the quality of another swarm intelligence optimization algorithm. This raises concerns about the neutrality of the evaluation.

Please include at least two additional evaluation criteria to provide a more comprehensive assessment of the algorithm’s performance. This will help ensure that the method performs well under diverse metrics and not only under those favorable to its design.

2. The paper compares the proposed method to four other state-of-the-art algorithms, which is commendable. However, the most recent benchmark cited dates back to 2018, with the earliest from 2003. To better demonstrate the value and potential of your approach, I recommend including comparisons with more recent methods from the past few years.

Minor:

1. The manuscript would benefit greatly from language editing. Many sentences are ambiguous or difficult to understand. Hiring a professional copy editor is strongly recommended.

2. Please define key concepts clearly when they are first introduced. For example, the term “overlapping impact of node influence” on page 11 needs further explanation for clarity.

3. On page 25, you state that the computational complexity of PDCSA is O(K ∙ ∙ ( ® + )). How does this compare with the computational complexity of the other benchmark algorithms? Including this comparison would provide helpful context for readers.

Reviewer #2: The article proposes an original algorithm called PDCSA (Parallel Discrete Crow Search Algorithm) to solve the influence maximization problem in social networks, focusing on the algorithm’s efficiency, computational complexity, and scalability. The novelty of the method and its experimentally demonstrated success are noteworthy. The combination of local and global search strategies effectively balances search space coverage and computational efficiency (pp. 18–21). All steps of the algorithm (especially Figure 2 and Algorithms 1–4) are clearly visualized and supported by a process flow diagram (pp. 16–19). Comparisons with state-of-the-art algorithms such as CELF, DBA, and DPSO show that PDCSA is both effective and fast (pp. 27–30). However, there are still several aspects of the study that require improvement.

1. The section concerning the theoretical analysis of the algorithm lacks depth, particularly in analytically explaining the factors that affect optimization performance. A more detailed mathematical analysis would be beneficial (pp. 22–23).

2. Although the references to the swarm intelligence literature are sufficient, the lack of any mention of GNN-based (Graph Neural Network) methods is a significant omission. GNNs have also proven to be effective in solving influence maximization problems in large-scale networks.

3. There are several English language issues throughout the manuscript. Some sentences reflect translation artifacts and should be revised:

o Sentences such as “The core objective... is to locate a set of nodes of size K.” (p. 16) should be rephrased in a more academic manner.

o Phrases like “From the existing research, many investigations... have been conducted.” (p. 12) are too general and need to be made more specific and original.

4. While experimental results are presented through figures (Figures 7–9), there is a lack of comparative tables displaying numerical data. Tables showing LIE values, runtime, and spread performance (e.g., “Table 3: LIE Value vs Time vs Seed Set Size”) should be included for greater clarity.

5. No link to the source code of the PDCSA algorithm is provided. The code (e.g., on GitHub) should be made publicly available, and this information should be clearly stated in the “Data Availability” section (pp. 6–7).

In conclusion, while the article presents a significant technical contribution and demonstrates strong experimental performance, it could make an even greater impact if certain deficiencies are addressed. In particular, English language editing, broader literature comparison, and clearer presentation of results via tables are needed.

6. PLOS authors have the option to publish the peer review history of their article (what does this mean? ). If published, this will include your full peer review and any attached files.

**Do you want your identity to be public for this peer review?** For information about this choice, including consent withdrawal, please see our Privacy Policy .

Reviewer #1: No

Reviewer #2: No

---

## [Decision Letter · Decision Letter 1]

16 Jul 2025

PDCSA: A parallel discrete crow search algorithm for influence maximization in social networks

PONE-D-24-54198R1

Dear Dr. Han,

We’re pleased to inform you that your manuscript has been judged scientifically suitable for publication and will be formally accepted for publication once it meets all outstanding technical requirements.

Kind regards,

Kuldeep Singh

Academic Editor

PLOS ONE

Additional Editor Comments (optional):

Reviewers' comments:

Reviewer's Responses to Questions

**Comments to the Author**

1. If the authors have adequately addressed your comments raised in a previous round of review and you feel that this manuscript is now acceptable for publication, you may indicate that here to bypass the “Comments to the Author” section, enter your conflict of interest statement in the “Confidential to Editor” section, and submit your "Accept" recommendation.

Reviewer #2: All comments have been addressed

2. Is the manuscript technically sound, and do the data support the conclusions?

Reviewer #2: (No Response)

3. Has the statistical analysis been performed appropriately and rigorously? 

Reviewer #2: Yes

4. Have the authors made all data underlying the findings in their manuscript fully available?

Reviewer #2: Yes

5. Is the manuscript presented in an intelligible fashion and written in standard English?

Reviewer #2: Yes

6. Review Comments to the Author

Reviewer #2: The authors have carried out a comprehensive revision in response to the reviewers' previous feedback. The revised version has been significantly improved in terms of both content and structure. The authors have added a theoretical framework explaining the factors affecting optimization performance (pp. 20–21). The balance between exploration and exploitation in the PDCSA algorithm has been detailed along with mathematical formulations. The revised manuscript is generally more fluent, and grammatical errors and translation issues in the Abstract, Introduction, and Conclusion sections have been largely resolved. Tables such as Table 4 and Table 5 have been added, clearly presenting the performance of PDCSA compared to other algorithms in terms of LIE, runtime, and influence spread metrics (pp. 25–28). The authors have also indicated that the PDCSA algorithm’s source code is accessible via GitHub, and the link has been included in the “Data Availability” section (p. 34). Overall, the authors have significantly improved the manuscript by addressing the reviewers' comments.

7. PLOS authors have the option to publish the peer review history of their article (what does this mean? ). If published, this will include your full peer review and any attached files.

**Do you want your identity to be public for this peer review?** For information about this choice, including consent withdrawal, please see our Privacy Policy .

Reviewer #2: No

---

## [Editor Report · Acceptance letter]

PONE-D-24-54198R1

PLOS ONE

Dear Dr. Han,

I'm pleased to inform you that your manuscript has been deemed suitable for publication in PLOS ONE. Congratulations! Your manuscript is now being handed over to our production team.

Kind regards,

on behalf of

Dr. Kuldeep Singh

Academic Editor

PLOS ONE